# Deep neural network-estimated electrocardiographic age as a mortality predictor

Emilly M. Lima[1,2,8], Antônio H. Ribeiro[3,4,8], Gabriela M. M. Paixão[1,2,8], Manoel Horta Ribeiro[5],
Marcelo M. Pinto-Filho [1,2], Paulo R. Gomes [1,2], Derick M. Oliveira[3], Ester C. Sabino [6], Bruce B. Duncan [7],
Luana Giatti[2], Sandhi M. Barreto[2], Wagner Meira Jr[3], Thomas B. Schön [4✉] & Antonio Luiz P. Ribeiro [1,2✉]

The electrocardiogram (ECG) is the most commonly used exam for the evaluation of cardiovascular diseases. Here we propose that the age predicted by artificial intelligence (AI) from the raw ECG (ECG-age) can be a measure of cardiovascular health. A deep neural network is trained to predict a patient's age from the 12-lead ECG in the CODE study cohort ($n = 1,558,415$ patients). On a 15% hold-out split, patients with ECG-age more than 8 years greater than the chronological age have a higher mortality rate (hazard ratio (HR) 1.79, $p < 0.001$), whereas those with ECG-age more than 8 years smaller, have a lower mortality rate (HR 0.78, $p < 0.001$). Similar results are obtained in the external cohorts ELSA-Brasil ($n = 14,236$) and SaMi-Trop ($n = 1,631$). Moreover, even for apparent normal ECGs, the predicted ECG-age gap from the chronological age remains a statistically significant risk predictor. These results show that the AI-enabled analysis of the ECG can add prognostic information.

[1] Telehealth Center, Hospital das Clínicas, Universidade Federal de Minas Gerais, Belo Horizonte, Brazil. [2] Faculdade de Medicina, Universidade Federal de Minas Gerais, Belo Horizonte, Brazil. [3] Departamento de Ciência da Computação, Universidade Federal de Minas Gerais, Belo Horizonte, Brazil. [4] Department of Information Technology, Uppsala University, Uppsala, Sweden. [5] École Polytechnique Fédérale de Lausanne, Lausanne, Switzerland. [6] Instituto de Medicina Tropical da Faculdade de Medicina da Universidade de São Paulo, São Paulo, Brazil. [7] Programa de Pós-Graduação em Epidemiologia and Hospital de Clínicas de Porto Alegre, Universidade Federal do Rio Grande do Sul, Porto Alegre, Brazil. [8] These authors contributed equally: Emilly M. Lima, Antônio H. Ribeiro, Gabriela M. M. Paixão. ✉email: thomas.schon@it.uu.se; antonio.ribeiro@ebserh.gov.br

The electrocardiogram (ECG) is the most commonly used exam for the screening and evaluation of cardiovascular diseases. Computerized, rule-based, ECG interpretation was developed to facilitate medical research and clinical practice. However, the limited accuracy of these methods has limited their application[1,2]. In this context, deep neural networks (DNNs) are a promising machine learning approach for the automated analysis of the ECG and have achieved unprecedented performance in initial studies[3,4].

DNNs present a paradigm shift from classical ECG automated analysis methods. Classical methods use signal processing techniques to extract the measurements, wavelengths, and detect abnormal beats from the ECG signal and then use the extracted information as input features to a classifier[5]. DNN-based ECG analysis, on the other hand, is based on an "end-to-end" approach, for which the raw signal is used as an input to the classifier, which learns to extract the features by itself[3,4].

Unlike the traditional methods, features learned by end-to-end ECG automated analysis methods do not necessarily have an interpretation rooted in electrocardiographic knowledge. If this paradigm introduces new challenges regarding model interpretability[6] and out-of-distribution robustness[7], it also introduces new possibilities when it comes to applications. Examples that go beyond traditional electrocardiography and have been achieved using end-to-end approaches include: predicting the risk of death from the ECG[8]; identifying patients who will develop atrial fibrillation from a previous ECG taken during sinus rhythm[9]; and screening for cardiac contractile dysfunction using only the 12-lead ECG[10]. This suggests that end-to-end models might be able to identify additional markers that, in turn, might be practical and useful tools in cardiovascular disease prediction.

In this context, we turn to the use of machine learning algorithms to infer age from ECG tracings[11,12]. Previous studies have shown that the age estimated from the ECG (the ECG-age) is related to cardiovascular health[11,12]: The ECG-age, calculated using a Bayesian model in 5-min ECGs, tended to be close to the chronological age in healthy non-athletes, whereas most subjects with risk factors or proven heart diseases had a predicted ECG-age that was higher than their chronological age[11]; in another study, patients with a DNN-predicted age that exceeded the chronologic age by 7 or more years presented a higher frequency of low ejection fraction, hypertension, and coronary disease[12]. More recently, ECG-derived age has also been related to vascular aging, measured by peripheral endothelial dysfunction[13].

Biological aging refers to the decline in tissue/organismal function, whereas chronological aging simply indicates the time passed since birth[14]. In normally aging individuals, chronological and biological ages are the same. Biological aging, however, is affected by lifestyle, environmental factors, inheritable and acquired conditions, and diseases. Accelerated biological aging points to the decline of tissue/organismal function at a faster rate than the average, and hence associated with the risk of a premature death[14]. Most available biomarkers of biological age measure a specific aspect of aging, like molecular and cellular biomarkers, and functional and structural vascular parameters[14]. Multiple exams and composite scores are often needed, adding cost, risk, and complexity to the evaluation. ECG exams are low-cost and widely available, being part of the routine evaluation of many patients in both primary and specialized care. Thus, if ECG can provide an accurate estimate of the biological age it can be potentially useful in clinical practise.

We build on the hypothesis that an AI model exposed to many ECG exams with the task of predicting the age might learn to capture, on average, how aging affects the ECG exam. Age is a risk factor for cardiac diseases that affects ECG measurements and the likelihood of having an ECG with a higher incidence of abnormalities[15,16]. Hence, here we study the possibility of using this AI predicted ECG-age as an indicator of cardiovascular health. We refer to this age predicted by the AI model from the raw 12-lead ECG tracing as predicted ECG-age, or just ECG-age, and, to contrast that, we refer to the patient age as chronological age.

In this paper, we demonstrate that this AI predicted ECG-age is a potentially useful tool in the assessment of the risk of death in the general population. We developed, in the CODE Study cohort[17], a DNN-based age-prediction model and assessed if the difference between predicted ECG-age and chronological age is a predictor of overall mortality. The model is validated in two external cohorts, ELSA-Brasil[18], of Brazilian public servants, and SaMi-Trop[19], of Chagas disease patients. Furthermore, we tested if the predictive value remains significant after controlling for the presence of cardiovascular risk factors and for subjects with normal ECGs. We sought to determine whether it can be used as a prognostic marker in the general population. Finally, we also undertook an exploratory analysis to investigate mechanisms that are involved in ECG-age prediction, looking at the main components used during the classification. This is done both by analyzing the model sensitivity to changes in the ECG signal and by the manual review of the ECGs and the corresponding predicted ECG-age by trained cardiologists.

## Results

**Deep neural network age-predictor model**. We used the CODE Study cohort[17] to develop a DNN capable of predicting the patient's age from the raw ECG tracing. The dataset consists of ECG records from 1,558,415 patients of 811 counties in the state of Minas Gerais (Brazil) collected by the Telehealth Network of Minas Gerais (TNMG). Patients were divided into 85–15% splits with the 85% split being used to develop the model (see Methods section).

The model is evaluated in three different cohorts, unseen by the DNN model during its development, the 15% hold-out split described above, which will be referred to as the CODE-15% cohort (with 218,169 participants), the ELSA-Brasil (with 14,236 participants), and the SaMi-Trop cohorts (with 1631 participants). Table 1 summarizes the baseline characteristics for each of the cohorts including median follow-up and number of events. Compared to the CODE-15% cohort, mean age, the prevalence of cardiovascular risk factors, and previous myocardial infarction were higher in both ELSA-Brasil and SaMi-Trop cohorts. The frequency of events was the highest in the SaMi-Trop cohort, composed of Chagas disease patients, many with chronic cardiomyopathy.

We used the DNN architecture known as the residual network[20] to perform the task. The architecture has been successfully used for ECG abnormality detection in previous work[3,4]. The DNN mean absolute error (MAE) in the age prediction task is 8.38 (with standard deviation, s.d., 7.00), 8.44 (s.d. 6.19), and 10.04 (s.d. 7.76) for the CODE-15%, ELSA-Brasil, SaMi-Trop, respectively. Figure 1 shows the relation between predicted ECG-age and chronological age for all the patients in the cohorts.

In the following sections, we try to establish the prognostic relevance of the ECG-age. We perform regression analyses that use the ECG-age as an input variable. In these analyses we always use the CODE-15% cohort for deriving the statistics and the ELSA-Brasil and SaMi-Trop for validating them.

**Electrocardiographic age as a mortality predictor**. We try to establish the relevance of ECG-age as a predictor of mortality. We

**Table 1 Baseline characteristics.**

|  | CODE-15% (n = 218,169) | ELSA-Brasil[b] (n = 14,263) | SaMi-Trop[c] (n = 1631) |
|---|---|---|---|
| *Characteristics[a]* |  |  |  |
| Sex, male, n (%) | 88,508 (41) | 6494 (46) | 550 (34) |
| Age (years), mean (s.d.) | 51 (20) | 52 (9) | 60 (13) |
| Hypertension, n (%) | 64,767 (30) | 5108 (36) | 593 (36) |
| Diabetes, n (%) | 13,720 (6) | 2830 (20) | 161 (10) |
| Smoking, n (%) | 13,645 (6) | 1882 (13) | 498 (31) |
| Previous myocardial infarction, n (%) | 1553 (0.7) | 258 (1.8) | 76 (5) |
| Follow-up(years), median (IQR) | 3.4 (2.1–5.0) | 9.8 (8.9–10.0) | 2.1 (2.0–2.2) |
| Events, n (%) | 8110 (3.7) | 617 (4.3) | 104 (6.4) |

The table summarizes the characteristics of the three cohorts analyzed in this study. It includes the baseline characteristics, the summary of follow-up time, and the number of events.
[a]Data are expressed as number (percentage) unless otherwise indicated.
[b]There are missing values in variables from the ELSA-Brasil cohort (hypertension, 13; diabetes, 3; smoking, 1; previous myocardial infarction, 7); valid percentages are reported.
[c]There are missing values in a variable from the SaMi-Trop cohort (smoking, 6); valid percentages are reported.

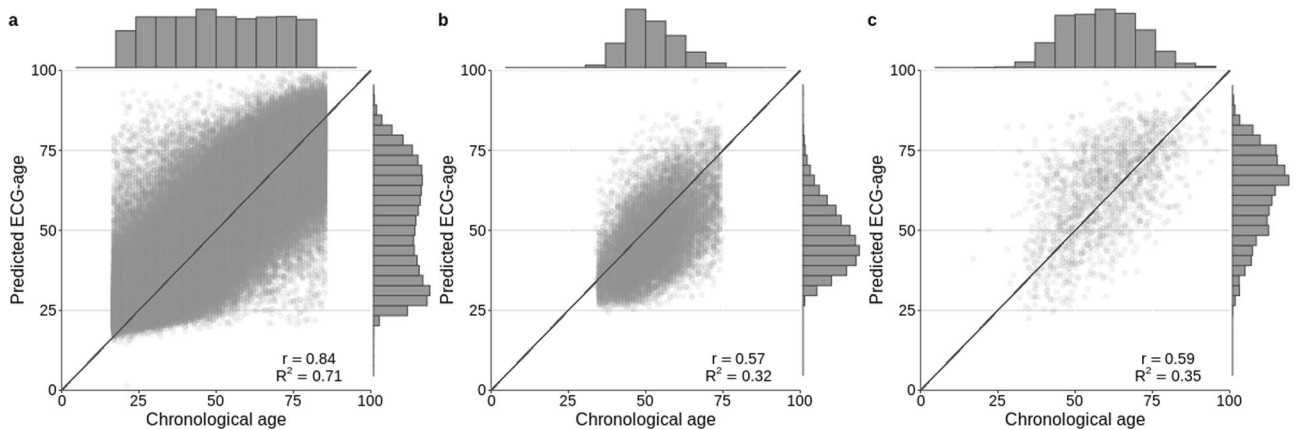

**Fig. 1 Chronological vs ECG-age.** The scatter plots display the relation between the predicted ECG-age and chronological age. The black line is the identity line. The lateral histograms show the distributions of predicted ECG-age and chronological age among patients of the cohorts. **a** CODE-15% cohort, **b** ELSA-Brasil cohort, **c** SaMi-Trop cohort. The mean predicted ECG-age was 52 (s.d. 19), 47 (s.d. 11), 63 (s.d. 14), for CODE-15%, ELSA-Brasil cohort and SaMi-Trop cohort, respectively. The $R^2$ (Pearson correlation) was 0.71 ($r = 0.84$) in the CODE-15%, 0.32 ($r = 0.57$) in ELSA-Brasil cohort and 0.35 ($r = 0.59$) in the SaMi-Trop cohort.

divided the patients into three groups, based on differences between predicted ECG-age and chronological age: (a) those with ECG-age more than 8 years greater than the chronological age; (b) those with ECG-age within a range of 8 years from their chronological age; and, (c) those with ECG-age more than 8 years smaller than the chronological age. The MAE in the CODE dataset is ~8 years, which motivates our choice for the thresholds used. That is, when the predicted ECG-age deviates from the chronological age by more than the mean deviation found in the derivation cohort we classify into group (a) if the deviation is positive, and into group (c), if it is negative. Experiments with alternative choices of threshold yield qualitatively similar results.

The risk of death for these three groups, expressed by their hazard ratios (HR), is shown in Table 2, together with the 95% confidence intervals (CI). We fit a Cox model, adjusted for age and sex, in the CODE-15% cohort. The adjusted survival curves for this model are presented in Fig. 2. This model indicates that participants with an estimated ECG-age of more than 8 years greater than the chronological age had higher mortality risk (HR 1.79, 95% CI 1.69–1.90; $p < 0.001$). On the other hand, those with an estimated ECG-age of more than 8 years smaller than the chronological age had a lower mortality risk (HR 0.78, 95% CI 0.74–0.83, $p < 0.001$). Results in the ELSA-Brasil cohort, were similar: with a higher mortality risk (HR 1.75, 95% CI 1.35–2.27; $p < 0.001$) for those with estimated ECG-age of more than 8 years greater than the chronological age; and a lower mortality rate (HR

0.74, 95% CI 0.63–0.88; $p < 0.001$) for those with ECG-age more than 8 years years less than the chronological age. In the SaMi-Trop cohort, patients with an ECG-age more than 8 years greater than the chronological age had a higher mortality risk (HR 2.42, 95% CI 1.53–3.83; $p < 0.001$); for ECG-age more than 8 years smaller than the chronological age, however, the observed decrease in mortality risk was not statistically significant (HR 0.89, 95% CI 0.52–1.54; $p = 0.68$)). Additional analysis also show that Cox model adjusted by sex and age presents a good performance in the prediction of 1-year mortality, with an area under the curve, AUC, of 0.80 (95% CI 0.79–0.81) for the CODE-15% cohort, 0.77 (95% CI 0.66–0.87) for the ELSA-Brasil, and 0.74 (95% CI 0.68–0.80) for the SaMi-Trop.

The importance of the ECG-age in predicting mortality remains also when we adjust the model for cardiovascular risk factors. Hazard ratios for models adjusted by different selections of variables cardiovascular risk factors are given in Table 2. In this analysis, we additionally adjusted the model for hypertension, diabetes mellitus, and smoking habits, but this did not yield significant differences in the results. As in the first case, all associations (except for ECG-age more than 8 years smaller than the chronological age in the Sami-Trop cohort) remained significant with little change in the adjusted HR. We also did additional adjustments for dyslipidemia (CODE-15% and ELSA-Brasil cohorts) and obesity (ELSA-Brasil), without changing significantly the magnitude of the observed association.

**Table 2 Risk of death.**

| | CODE-15% (n = 218,169) | | ELSA-Brasil (n = 14,263) | | SaMi-Trop (n = 1,631) | |
|---|---|---|---|---|---|---|
| | HR (CI 95%) | p value | HR (CI 95%) | p value | HR (CI 95%) | p value |
| *Adjusted by age and sex* | | | | | | |
| ECG-age < age−8 years | 0.78 (0.74–0.83) | <0.001 | 0.74 (0.63–0.88) | <0.001 | 0.89 (0.52–1.54) | 0.681 |
| ECG-age > age+8 years | 1.79 (1.69–1.90) | <0.001 | 1.75 (1.35–2.27) | <0.001 | 2.42 (1.53–3.83) | <0.001 |
| *Adjusted by age, sex, hypertension, diabetes mellitus, and smoking* | | | | | | |
| ECG-age < age−8 years | 0.78 (0.74–0.83) | <0.001 | 0.82 (0.69–0.98) | 0.030 | 0.90 (0.52–1.55) | 0.702 |
| ECG-age > age+8 years | 1.79 (1.68–1.89) | <0.001 | 1.56 (1.20–2.03) | <0.001 | 2.48 (1.56–3.94) | <0.001 |
| *Adjusted by age, sex, hypertension, diabetes mellitus, smoking, and dyslipidemia* | | | | | | |
| ECG-age < age−8 years | 0.78 (0.74–0.83) | <0.001 | 0.82 (0.69–0.98) | 0.030 | Not available | |
| ECG-age > age+8 years | 1.78 (1.68–1.89) | <0.001 | 1.56 (1.20–2.03) | <0.001 | Not available | |
| *Adjusted by age, sex, hypertension, diabetes mellitus, smoking, dyslipidemia and obesity* | | | | | | |
| ECG-age < age−8 years | Not available | | 0.82 (0.69–0.98) | 0.030 | Not available | |
| ECG-age > age+8 years | Not available | | 1.57 (1.21–2.04) | <0.001 | Not available | |

The table displays the hazards ratios (HR) when the difference between ECG-age and chronological age are larger than 8 years (either positive or negative). The HR summarizes the Cox regression models obtained for overall mortality. The models were adjusted by different selection of variables (including age, sex, and cardiovascular risk factors).
The number of death events was n = 8,118 for CODE-15%, n = 617 for ELSA-Brasil, and n = 104 for SaMi-Trop. When the ECG-age is more than 8 years smaller than the chronological age n = 1,861, n = 239, and n = 19, respectively, for the CODE-15%, ELSA-Brasil, and SaMi-Trop cohorts. When the ECG-age is more than 8 years greater than the chronological age n = 1,675, n = 69, and n = 41, respectively.

Supplementary Table 1 presents baseline characteristics for each of the cohorts by ECG-age groups. We find that in all three cohorts, the group of patients with an ECG-age of more than 8 years greater than the chronological age have the lowest average chronological age (CODE-15%: 42.2; ELSA-Brasil: 48.5; SaMi-Trop: 54, $p < 0.001$ for $t$-tests comparing average ages with the other ECG-age strata). Although seemingly contradictory, this is in accordance with the hypothesis that ECG-age is indeed a predictor of mortality. Since patients that have an ECG-age higher than their chronological age are more likely to die, older patients whose ECG-age is higher than their chronological age are not likely to be a part of the sample we are analyzing.

**Electrocardiographic age as a mortality predictor in apparently normal ECGs.** Table 3 describes conventional ECG measurements for the participants in the three groups described above - i.e., (a) patients with predicted ECG-age more than 8 years greater than their chronological age; (b) more than 8 years smaller than their chronological age; and, (c) within a range of 8 years from their chronological age. Although in the CODE-15% cohort statistically significant differences can be seen for all measurements ($p < 0.001$ for all), these numbers do not yield a clinically significant difference. From a clinical perspective, these measurements can be considered remarkably similar to each other. In the ELSA-Brasil cohort, measurements were also numerically similar with a statistically significant difference obtained only for heart rate ($p < 0.001$) and QTc interval ($p < 0.001$).

To further evaluate whether the predicted ECG-age by the DNN was related to traditional electrocardiographic abnormalities, we performed an additional analysis, now restricted to normal ECGs from the CODE-15% and ELSA-Brasil cohorts. Which have, respectively, 80679 and 7691 participants with normal ECGs. We did not perform this analysis in the SaMi-Trop because most patients had ECG abnormalities related to Chagas disease. What was considered as normal ECG is defined in Methods section. An analysis with a Cox model restricted to the normal ECG was performed and the obtained hazard ratios are displayed in Table 4. The same parameters of the analysis in Table 2 are used. In the model adjusted by age and sex, ECG-age more than 8 years greater than the chronological age remained a statistically significant predictor of death risk in both cohorts (HR 1.53, 95% CI 1.30–1.80, $p < 0.001$ in CODE-15% and HR 1.63, 95% CI 1.00–2.66 $p = 0.050$ in ELSA-Brasil). On the other hand, ECG-age more than 8 years smaller than the chronological age remained associated with reduced risk of mortality in the CODE-15% (HR 0.66, 95% CI 0.57–0.76 $p < 0.001$) but was not statistically significant in the ELSA-Brasil cohort (HR 0.91, 95% CI 0.68–1.21 $p = 0.502$).

The results for models additionally adjusted for cardiovascular risk factors is also displayed in Table 4. After the adjustment, ECG-age more than 8 years greater than the chronological age was associated with increased risk of mortality in CODE-15% cohort (HR 1.52, 95% CI 1.29–1.79, $p = 0.015$), but not in ELSA-Brasil (HR 1.49, 95% CI 0.91–2.43, $p = 0.114$). This was also true for an ECG-age more than 8 years smaller than the chronological age. Risk was significantly decreased in CODE-15% cohort (HR 0.66, 95% CI 0.57–0.76, $p < 0.001$) but not in ELSA-Brasil (HR 1.00, 95% CI 0.75–1.33, $p = 0.990$). Which might be justified by the lack of statistical power due to the small number of deaths in this group for the ELSA-Brasil cohort ($n = 19$). Additional adjustments for dyslipidemia (CODE-15% and ELSA-Brasil cohorts) and obesity (ELSA-Brasil) do not qualitatively change the results.

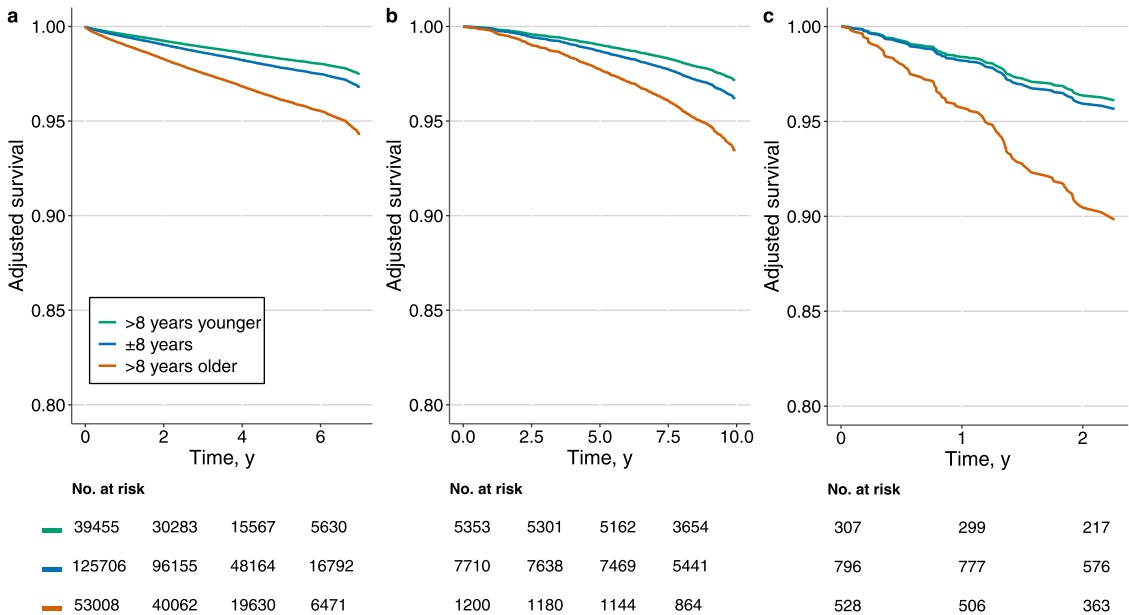

**Fig. 2 Adjusted survival curves.** The plots display the survival curves for the different cohorts. **a** CODE-15% cohort, **b** ELSA-Brasil cohort, **c** SaMi-Trop cohort. The curves are computed from the age and sex-adjusted Cox proportional model for all-cause mortality. Three groups of patients are taken into consideration: those with ECG-age more than 8 years greater than the chronological age (denoted by: ">8 years older"); those with ECG-age within a range of 8 years from their chronological age (denoted by: "±8 years"); and, those with ECG-age more than 8 years smaller than the chronological age (denoted by: ">8 years younger").

**Table 3 ECG measurements.**

| | CODE-15% ($n = 80,679$) | | | | ELSA-Brasil ($n = 7,691$) | | | |
|---|---|---|---|---|---|---|---|---|
| | ± 8 years | >8 years younger | >8 years older | p value | ± 8 years | >8 years younger | >8 years older | p value |
| Heart rate (bpm) | 70 (63–78) | 70 (62–79) | 71 (64–79) | <0.001 | 66 (61–72) | 64 (59–71) | 69 (63–75) | <0.001 |
| P duration (ms) | 106 (100–114) | 108 (100–116) | 108 (100–116) | <0.001 | 108 (102–116) | 110 (102–116) | 108 (100–116) | 0.558 |
| QRS axis | 47 (27–65) | 45 (25–62) | 43 (24–60) | <0.001 | 44 (21–60) | 43 (20–61) | 44 (19–61) | 0.737 |
| QRS duration (ms) | 90 (84–96) | 90 (84–96) | 92 (84–98) | <0.001 | 86 (80–92) | 86 (82–92) | 86 (80–90) | 0.068 |
| Average RR interval (ms) | 845 (757–942) | 845 (750–950) | 837 (750–932) | <0.001 | – | – | – | – |
| QTc (ms) | 411 (400–424) | 413 (401–425) | 413 (401–425) | <0.001 | 416 (405–427) | 414 (403–426) | 418 (406–429) | <0.001 |

The table displays the median, and ("under parentesis") the interquartile range, for the ECG measurements. It considers three groups of patients: those with ECG-age more than 8 years greater than the chronological age (denoted by: ">8 years older"); those with ECG-age within a range of 8 years from their chronological age (denoted by: "±8 years"); and, those with ECG-age more than 8 years smaller than the chronological age (denoted by: ">8 years younger"). Statistical comparison of the medians is made through Kruskal–Wallis two-sided test.

**Electrocardiographic age and cardiovascular risk factors**. Figure 3a represents which cardiovascular risk factors were most likely associated with a predicted ECG-age more than 8 years greater than the chronological age considering all ECGs from ELSA-Brasil cohort. After logistic regression adjusted for age and sex, hypertension, diabetes, smoking, and obesity remained significantly associated with an increased odds of having an ECG-age more than 8 years greater than the chronological age. In Fig. 3b the same model was applied only to participants with a normal ECG. Hypertension, diabetes, and smoking were significantly associated with a predicted ECG-age more than 8 years greater than the chronological age.

**Interpretability and time and frequency domain saliency maps**. To assess whether ECG-age captures signals that can be interpreted by cardiologists, we conducted an additional experiment. We paired ECG-ages of subjects with the same chronological age, but where one of them had an ECG-age more than 8 years greater

than their chronological age and the other more than 8 years smaller than their chronological age. Then, three medical doctors were asked to independently determine, for each pair, which ECG tracing was associated with the subject with higher ECG-age. Analyzing doctor's assessments of 134 pairs of traces, aggregated through majority voting, we found that they were not significantly better than random ($\chi^2 = 3.0$, $p = 0.12$). We provide detailed results in Supplementary Table 2. Throughout the experiment, doctors were given feedback about their predictions (in Stage 2), this did not increase their accuracy in the subsequent stage. In fact, they performed worse in Stage 3 (accuracy = 45.5%), after the feedback, than in Stage 1 (accuracy = 64.4%), before the feedback, or in Stage 2 (accuracy = 62.2%), during the feedback.

Additionally, we randomly generated 50 pairs of normal ECG tracings, with saliency maps[21] highlighting the regions in the ECG tracing that have the highest impact in the predicted ECG-age (see Methods section). Supplementary Fig. 1 provides some

**Table 4 Hazard Ratios for normal ECGs.**

| | CODE-15% (n = 80,679) | | ELSA-Brasil (n = 7,691) | |
|---|---|---|---|---|
| | HR (CI 95%) | p value | HR (CI 95%) | p value |
| *Adjusted by age and sex* | | | | |
| ECG-age < age−8 years | 0.66 (0.57–0.76) | <0.001 | 0.91 (0.68–1.21) | 0.502 |
| ECG-age > age+8 years | 1.53 (1.30–1.80) | <0.001 | 1.63 (1.00–2.66) | 0.050 |
| *Adjusted by age, sex, hypertension, diabetes mellitus, and smoking* | | | | |
| ECG-age < age−8 years | 0.66 (0.57–0.76) | <0.001 | 1.00 (0.75–1.33) | 0.990 |
| ECG-age > age+8 years | 1.52 (1.29–1.79) | <0.001 | 1.49 (0.91–2.43) | 0.114 |
| *Adjusted by age, sex, hypertension, diabetes mellitus, smoking, and dyslipidemia* | | | | |
| ECG-age < age−8 years | 0.66 (0.57–0.76) | <0.001 | 1.00 (0.75–1.33) | 0.990 |
| ECG-age > age+8 years | 1.52 (1.29–1.79) | <0.001 | 1.49 (0.91–2.43) | 0.114 |
| *Adjusted by age, sex, hypertension, diabetes mellitus, smoking, dyslipidemia, and obesity* | | | | |
| ECG-age < age−8 years | Not available | | 1.00 (0.75–1.33) | 0.992 |
| ECG-age > age+8 years | | | 1.42 (0.86–2.35) | 0.171 |

The table displays, for patients with a normal ECG, the hazard ratios (HR) according to the differences between ECG-age and chronological age. The HR summarizes the Cox regression models obtained for overall mortality. The models were adjusted by different selection of variables (including age, sex, and cardiovascular risk factors).
The number of events was n = 1074 for CODE-15% and n = 228 for ELSA-Brasil. The number of events when ECG-age is more than 8 years smaller than the chronological age there were, n = 249 and n = 105 for CODE-15% and ELSA-Brasil, respectively. Considering ECG-age is more than 8 years greater than the chronological age there were n = 203 and n = 19 events for CODE-15% and ELSA-Brasil, respectively.

illustrative examples. We asked the same set of three medical doctors to qualitatively analyze which sections of the ECG were being frequently highlighted by the visualization algorithm. Doctors independently suggested that low-frequency components, as P and T waves, were disproportionately highlighted.

We also generate saliency maps in the frequency domain giving the relative importance of each frequency component for the final prediction (see Methods section). Supplementary Figure 2 shows the median and interquartile range from this analysis for 100 normal exams in each cohort. The analysis suggests the frequency component between 8 and 15 Hz of the ECG spectrum are the ones that most contribute to the model prediction.

**A fine-grained analysis of the relation between ECG-age and mortality.** In most of the discussion in this paper we use a hard threshold of 8 years old between the predicted ECG-age and the chronological age to separate the patients into different risk groups. In Supplementary Fig. 4, we present an alternative setup where we split the patients into five quintiles of the difference of predicted ECG-age and chronological age and show adjusted survival, hazard ratios, and 95% confidence intervals for these groups. The results indicate that the groups where predicted ECG-age is lower than the chronological age (Q1 and Q2) had a lower risk of mortality and the groups Q4 and Q5 had a higher risk of mortality. Both when all exams are considered and when only normal exams are considered.

## Discussion

In this paper, we use a data-driven approach to obtain a model that predicts age from the raw ECG tracing. By having the chronological age of the person as the prediction target, we expect the trained model to learn to capture, on average, how aging affects the ECG exam. Indeed, having a predicted ECG-age higher than one's actual age is an indication that the exam is similar to those of older people, who have a higher associated cardiovascular risk and are more likely to die from cardiovascular diseases. We show that classical cardiovascular risk factors are associated with having an ECG-age more than 8 years greater than the chronological age. For some risk factors, such as hypertension, diabetes mellitus, and smoking, the association remains even when only normal ECGs were considered (cf. Fig. 3). Moreover, this study shows, in three different cohorts, that the difference

between the ECG-age and the chronological age can be used as a marker of the risk of death.

From a clinical perspective, ECG-age may present itself partly as a natural summary index of ECG changes and abnormalities accumulated during the life course of each subject. ECG tracings are affected by a large number of factors and mechanisms and, while summarizing them in a single number is a huge oversimplification, it can still be useful. It transmits the idea of cardiovascular risk in a language that does not require medical expertize and can be understood by patients and other professionals without medical training. In the literature, an AI-based model that predicts the probability of 1-year mortality have been recently proposed[8] and could also play a similar role. Nonetheless, reporting the ECG-age seems more intuitive from a patient perspective and, probably, easier to be used in clinical practice.

The analyses suggest that ECG-age is capable of capturing more than traditional ECG abnormalities or underlying conditions. Over-estimation of ECG-age was significantly associated with death after controlling for age and sex, cardiovascular risk factors and, even, when calculated only for subjects with normal ECGs. In the case of normal ECGs, this association was significant in CODE-15% but not in the ELSA-Brasil cohort. This might be explained by the small number of deaths in the ELSA-Brasil cohort or by the poorer annotation of risk factors in the CODE-15% study, in which this information is self-reported and obtained during the clinical activity. Moreover, ECG measurements were also not meaningfully different in the groups with predicted ECG-age more than 8 years greater than, more than 8 years smaller than, and within a range of 8 years from their chronological age.

Since the maintenance of a normal ECG status over time is associated with a low risk of cardiovascular diseases in a dose-response relationship[22], we hypothesize that the DNN might be able to identify subtle abnormalities that are not being currently identified in traditional analysis. This could help justify the capacity of evaluating the risk even for apparently normal ECGs. The lack of capability of trained doctors to distinguish between pairs of normal ECGs of the same age but different ECG-age (see Supplementary Table 2) also supports this hypothesis.

The advance of interpretable machine learning algorithms[23] might make it possible to leverage the features used by these models into clinical practice. Our initial insights on the mechanisms used for the estimation of ECG-age - and its

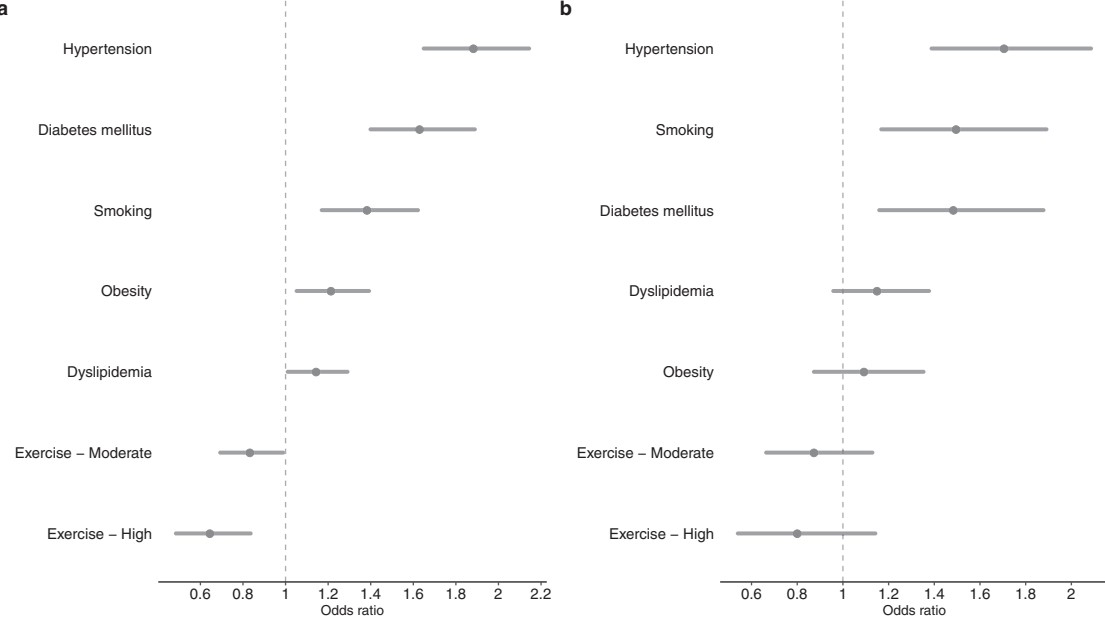

**Fig. 3 Adjusted odds ratios (ORs) for the ELSA-Brasil cohort.** The figure shows the adjusted ORs of the ECG-age being more than 8 years greater than the chronological age for risk factors. **a** All patients; and, **b** only for patients with normal ECG. The dots represent the adjusted ORs (by age and sex) and the horizontal lines represent the corresponding 95% CIs. HAS hypertension, DM diabetes mellitus, DLP dyslipidemia.

prognostic value - suggest that low-frequency components of the ECG, usually associated with P and T waves, might play an important role although these considerations would deserve a specific and more detailed investigation.

Despite being part of the routine evaluation of many patients in both primary and specialized care, the role of ECG exams are low-cost and widely available, the role in cardiovascular disease prediction and, hence, prevention is not as clear. Its prognostic impact has been explored in previous publications[24,25], nonetheless, the available methods are not widely adopted as a screening tool for individuals free of cardiovascular disease[26]. Our study is a further step towards a more practical use of the ECG in prognostic evaluation, considering that ECG-age can be a marker of the biological age of the cardiovascular system, or "cardiovascular age". This concept was introduced in previous studies[27,28] with the purpose of improving risk communication and patient adherence to proposed interventions. There, however, the value does not provide additional information to what the calculated risk already informs (since it is calculated based only on them). Doctors often struggle in decision making regarding treatment for primary prevention of cardiovascular diseases in intermediate risk patients. Identifying new risk modifiers that can potentially improve risk prediction in this population (either by a positive net reclassification index or by derivation of a new predictive model) is paramount. This is specially true if such a marker is derived from an inexpensive and widely available tool such as the ECG. The analysis presented here shows that the ECG-age can inform on risk that is not accounted for in traditional cardiovascular risk factors. And, in this sense, the concept can go beyond the concept of "cardiovascular age" proposed in previous studies[27,28].

Our work is perhaps best understood in the context of its limitations. The use of end-to-end DNN models is central to this work and yielded interesting findings (such as the possibility of predicting mortality even for apparent normal ECGs). Nonetheless, the complexity of these models makes it hard to fully interpret the results. Our exploratory analysis included sensitivity analysis both in the time and frequency domain and the analysis and review of more than 100 ECGs by trained cardiologists.

While it did provide some insight on what is being detected by the model, it is far from sufficient to completely explain the findings. Furthermore, while our study demonstrates the potential clinical utility of the ECG-age in individual risk prediction, further studies are desired to evaluate its incorporation in the clinical practise, including its use in addition to established risk calculators for primary prevention of cardiovascular diseases.

Here, we present the mortality risk prediction as a downstream task. That is, a model that was trained for predicting the patient's age is later used for a different task: that of mortality prediction. This shows the model is useful in scenarios that it has not been explicitly trained on and when used in a simple linear cox model it can help separate patients in different risk groups. Nonetheless, one possible limitation of this analysis is that the relations considered in this second step are only the linear ones. Hence, taking into consideration nonlinear relations in this second step could possibly modify the observed relationship.

To conclude, the predicted ECG-age may reflect biological age and it is a promising tool for risk prediction of overall mortality. It summarizes the information from the ECG in a single index with a clear interpretation for the patient. Data for training these models are also easy to obtain: while producing large datasets fully annotated with electrocardiographic abnormalities requires many hours of work by trained physicians, self-reported age is usually easy information to come by. Finally, the ability to predict mortality even for normal ECGs suggests that there might still be subtle electrocardiographic markers and abnormalities that are of interest and are not being captured in traditional analysis and the models presented here might be a useful tool in trying to find them.

## Methods

**Ethics declarations**. This study complies with all relevant ethical regulations. CODE Study was approved by the Research Ethics Committee of the Universidade Federal de Minas Gerais, protocol 49368496317.7.0000.5149. Since this is a secondary analysis of anonymized data stored in the TNMG, informed consent was not required by the Research Ethics Committee for the present study. ELSA-Brasil was approved by the Research Ethics Committees of the participating institutions and by the National Committee for Research Ethics (CONEP 976/2006) of the Ministry of Health. Sami-Trop study was approved by the Brazilian National

Institutional Review Board (CONEP), No. 179.685/2012. In both investigations, all human subjects were adults who gave written informed consent. All researchers who deal with datasets signed terms of confidentiality and data utilization.

**The CODE cohort.** Clinical Outcomes in Digital Electrocardiography (CODE) study[17] was developed with the database of digital ECG exams of the TeleHealth Network of Minas Gerais (TNMG)[29,30], Brazil, linked to the public databases of the Mortality and Hospitalization Information Systems. It was expected that the consolidated database would be useful for multiple purposes, including the evaluation of the epidemiological and prognostic significance of ECG findings[31] and the development of new methods of automatic classification of ECG abnormalities[3], using both conventional statistical methods and new machine learning techniques.

Patients over 16 years old with a valid ECG performed from 2010 to 2017 were included. Clinical data were self-reported. A hierarchical free-text machine learning algorithm recognized specific ECG diagnoses from cardiologist reports. The Glasgow ECG Analysis Program provided Minnesota Codes and automatic diagnostic statements. For the CODE database, the presence of a specific electrocardiographic diagnosis was considered automatically when there was an agreement between the diagnosis extracted from the cardiologist report and the automatic report from Glasgow Diagnostic Statements or Minnesota code. In cases where there were discordances between medical reports and one of the automatic programs, a manual revision was done by trained cardiologists[17].

The electronic cohort was obtained linking data from the ECG exams (name, sex, date of birth, and city of residence) and those from the national mortality information system, using standard probabilistic linkage methods (FRIL: Fine-grained record linkage software, v.2.1.5, Atlanta, GA). After the linkage, the data was anonymized for storage[17].

From a dataset of 2,470,424 ECGs, 1,773,689 patients were identified. After excluding the ECGs with technical problems and patients under 16 years old, a total of 1,558,415 patients were included for analyses. The mean age was 51.6 [s.d.17.6] years with 40.2% male. The overall mortality rate was 3.34% in a mean follow-up of 3.7 years[17].

The model was also evaluated in two established cohorts, the São Paulo-Minas Gerais Tropical Medicine Research Center (SaMi-Trop)[19] of Chagas disease patients and the Longitudinal Study of Adult Health (ELSA-Brasil)[18], of Brazilian public servants, in which raw ECG tracings from the baseline and follow-up with total mortality as the end-point are available. These cohorts are described next.

**The CODE-15% cohort.** The CODE-15% is a subset of the CODE cohort. The CODE cohort was divided into 85-15% splits, with the 85% split being used for developing the model and 15% hold-out being the one used in subsequent analyses and referred to as CODE-15%. This hold-out set is obtained by a stratified sampling procedure, where the stratification is made with respect to the patients age. The procedure is illustrated in Supplementary Fig. 5. Given all the exams from the original CODE cohort, we group the exams by the age of the patient at the time of the examination. One group for each age ranging from 16 to 85 years, i.e. a total of 70 uniformly spaced age groups. The CODE-15% cohort is obtained by picking the same number of exams (~3100) at random from each age group. The result is an, approximately, uniform age distribution from 16 years to 85 years. Only the first patient exam is considered in all the analysis with this cohort and the remaining exams from the patients are removed from the remaining data and not used in the analysis. We do not sample from patients older than 85 or younger than 16 years, which do not appear in the CODE-15% cohort.

**The ELSA-Brasil cohort.** ELSA-Brasil is a cohort study that aims to investigate the development of chronic diseases, primarily diabetes and cardiovascular diseases, over a long-term follow-up[32,33]. All active or retired employees of the six institutions (and, in a few instances, also of related educational or health institutions) from six Brazilian capitals, of both sexes, and with ages between 35 and 74 years, were eligible for the study. Exclusion criteria were severe cognitive or communication impairment, intention to quit work at the institution in the near future for reasons not related to retirement, and, if retired, residence outside the corresponding metropolitan area. Women with current or recent pregnancy were rescheduled so that the first interview could take place ≥4 months after delivery. A total of 15,105 participants were enrolled, 6887 men and 8218 women, thus giving reasonably large numbers for sex-specific analyses. Baseline assessment (2008–10) included detailed interviews and measurements to assess social and biological determinants of health, as well as various clinical and subclinical conditions related to diabetes, cardiovascular diseases, and mental health. A second and third visit of interviews and examinations were done (2012–14 and 2017–2019) to enrich the assessment of cohort exposures and to detect initial incident events. Annual surveillance has been conducted since 2009 for the ascertainment of incident events. Biological samples (sera, plasma, urine, and DNA) obtained at both visits have been placed in long-term storage. In a mean of 9.36 years of follow-up, 14,263 (94,5%) participants were followed, until 01/01/2020, 617 (4.3%) died and 842 (5.6%) were lost to follow-up.

**The SaMi-Trop cohort.** SaMi-Trop is an NIH-funded prospective cohort of 1959 patients with chronic Chagas cardiomyopathy to evaluate whether a clinical prediction rule based on ECG, brain natriuretic peptide (BNP) levels, and other biomarkers can be useful in clinical practice[19,34]. The study is being conducted in 21 municipalities of the northern part of Minas Gerais State in Brazil with at least 2 years of follow-up, including one visit at baseline and another at 24 months. Eligible patients were selected based on the ECG results performed in 2011–2012 by the Telehealth Network of Minas Gerais, which from now on will be called index ECG. Only patients who fulfilled all of the following inclusion criteria were selected: (1) self-reported Chagas disease; (2) an index ECG reported as abnormal and (3) aged 19 years or more. The exclusion criteria included pregnancy or breastfeeding, and any life-threatening disease with an ominous prognosis that suggested a life expectancy of <2 years. The baseline evaluation included a collection of sociodemographic information, social determinants of health, health-related behaviors, comorbidities, medicines in use, history of previous treatment for Chagas disease, functional class, quality of life, blood sample collection, and ECG. Patients were mostly female, aged 50–74 years, with low family income and educational level, with known Chagas disease for >10 years; 46% presented with functional class > II. Previous use of benznidazole was reported by 25.2% and permanent use of pacemaker by 6.2%. Almost half of the patients presented with high blood cholesterol and hypertension and one-third of them had diabetes mellitus. N-terminal of the prohormone BNP (NT- ProBNP) level was >300 pg/mL in 30% of the sample. Clinical and laboratory markers predictive of severe and progressive Chagas disease were identified as high NT-ProBNP levels, as well as symptoms of advanced heart failure[34]. During a mean follow-up of 2.09 years, 1631 patients were being followed until the 2nd visit. In total, 104 (6.4%) died and 328 (16.7%) were lost to follow-up.

**Electrocardiographic and clinical definitions in CODE and ELSA-Brasil.** An ECG was considered "normal" in the CODE cohort according to conventional clinical reporting and by having automatic measurements by the Glasgow software within the normal range. In the ELSA-Brasil and Sami-Trop cohorts, ECGs were codified by the Minnesota code[18,35] with manual review of a trained cardiologist. Those with no major or minor abnormalities according to the criteria were considered normal.

All clinical risk factors included in the CODE cohort were self-reported in a clinical standardized questionnaire. Hypertension, diabetes, and dyslipidemia were also considered if informed use of antihypertensives, oral hypoglycemic agents or insulin, statins or fibrates; respectively. In the Sami-Trop cohort, the risk factors were also self-reported in a baseline interview. In the ELSA-Brasil study, hypertension was defined as systolic blood pressure ≥140 mmHg or diastolic blood pressure ≥90 mmHg, or verified treatment with anti-hypertensive medication during the past 2 weeks; diabetes mellitus as a report of a previous diagnosis of diabetes, or the use of medication for diabetes, or meeting a diagnostic value for diabetes according to one of the following tests: fasting or 2-h plasma glucose obtained during a 75-g oral glucose tolerance test or HbA1C; dyslipidemia as either a total cholesterol ≥240 mg/dl, LDL cholesterol ≥160 mg/dl, HDL cholesterol <40 mg/dl or triglycerides ≥150 mg/dl; obesity as BMI ≥ 30 kg/m$^2$ and smoking by participants self-report.

**The model.** Exams from patients in the CODE cohort that were not included in the hold-out set CODE-15% (see section above) were used to develop a convolutional DNN to predict age. This split contains 85% of the patients and was further divided into 80-5% splits: being the first used to learn the neural network weights, and the samples from the 5% remaining patients used for comparing design choices and adjusting optimization parameters.

The 5% validation split for is obtained by using a stratified sampling procedure. The procedure is illustrated in Supplementary Fig. 4 and is similar to the one used for generating the CODE-15% cohort. Given all the patients that are not in the CODE-15% cohort, we group their exams by the age of the patient at the moment it was taken. One group for each age ranging from 16 to 85 years. The 5% validation set is then obtained by sampling the same number of exams (~1600) at random from each age group. As for the CODE-15% cohort, such a procedure aims to guarantee an, approximately, uniform age distribution in the validation set, by picking the same number of patient exams for equally spaced 1-year intervals. The training dataset is composed of all exams from the 80% remaining patients. The training dataset has an unbalanced distribution of ages and, to correct for it during the training procedure, we weight the exam records inversely proportional to the frequency of patients with that given age.

The architecture and the set of hyperparameters are described next and are similar to a previous study[3], for which the DNN was trained to detect 6 types of ECG abnormalities (considered representative of both rhythmic and morphologic ECG abnormalities) on the same dataset. The results with this choice of hyperparameters were considered satisfactory and no further hyperparameter search was performed.

We used a convolutional neural network similar to the residual network proposed for image classification[20], but adapted to unidimensional signals. This architecture allows deep neural networks to be efficiently trained by including skip connections. We have adopted the modification in the residual block proposed by He et al.[36].

All ECG recordings, which have between 7 and 10 s of duration and are sampled at frequencies ranging from 300 to 1000 Hz, are re-sampled to 400 Hz and zero-padded, resulting in signals of fixed length (4096 samples), which are fed to the neural network. The output is the age predicted for that given exam.

The network consists of a convolutional layer followed by five residual blocks with two convolutional layers per block. The output of each convolutional layer is rescaled using batch normalization[37] and fed into a rectified linear activation unit ReLU. Dropout[38] is applied after the nonlinearity. The convolutional layers have filter length 17, starting with 4096 samples and 64 filters for the first layer and residual block and increasing the number of filters by 64 and subsampling by a factor of 4 every residual block. Max Pooling[39] and convolutional layers with filter length 1 are included in the skip connections to make the dimensions match those from the signals in the main branch.

The weighted mean square error is minimized using Adam optimizer[40] with default parameters and a learning rate of 0.001. The learning rate is reduced by a factor of 10 whenever the validation loss does not present any improvement for seven consecutive epochs. The neural network weights were initialized sampling from a normal random variable scaled as in He et al.[41] and the bias was initialized with zeros. The training runs for 70 epochs with the final model being the one with the best validation results during the optimization process.

**Cardiologist assessment of ECG-age from the tracings**. To assess whether ECG-age was capturing ECG changes that are recognizable to medical doctors, we conducted an additional experiment asking three experienced medical doctors to identify, in paired ECGs, ECG tracings associated with having higher ECG-age. All ECGs considered were normal ECGs from the CODE cohort. Within each pair of equal chronological age and sex, one individual had an ECG-age more than 8 years greater than their chronological age and the other had an ECG-age more than 8 years smaller than their chronological age. We included one pair of male and one pair of female patients for each age between 16 and 85 (whenever possible), totaling 134 pairs. At the edges of our age-range, it was not always possible to have an ECG tracing with ECG-age more than 8 years smaller than the chronological age paired with a tracing with ECG-age more than 8 years smaller than the chronological age, and, in these situations, we use tracings associated with ECG-ages within the 8 years range of the patient's chronological age. The experiment was divided into three stages where doctors annotated 44, 45, and 45 pairs of ECGs tracings respectively. In stages 1 and 3, doctors were not given the answer after accomplishing the task, and in stage 2 they were. The idea behind this distinction is to see whether doctors would fare any better after a round with explicit feedback on their performance.

**Saliency maps in the time and frequency domain**. We performed an analysis to assess the relative importance of different segments of the ECG trace in the age prediction. The results are displayed in Supplementary Fig. 1 and the relative size of the blue disks in the image might be interpreted as the relative importance of each point to the output prediction (at least in terms of the linearized local analysis). Similar approaches have been pursued in the interpretation of other DNN-based ECG predictors[8,42]. Here we use a rather straightforward procedure for generating the saliency maps[21]: the raw ECG tracing is fed to the deep neural network and the ECG-age is computed. Using backpropagation we compute the derivative of the ECG-age with respect to each point. We then generate transparent blue disks in the same plot as the ECG tracing, where the size of these disks is proportional to the magnitude of the derivative in this point. This procedure results in the saliency map displayed in Supplementary Fig. 1.

In Supplementary Fig. 2, we show a similar analysis, but now in the frequency domain. We take the discrete Fourier transform of the gradients computed as described above. We do that for 100 ECG exams, sampled at random, from the ECG exams classified as normal in each of the three different cohorts (CODE-15%, ELSA-Brasil, and SaMi-Trop) and show the median and interquartile range in the Figure.

**Statistical analysis**. To assess the performance of the DNN model in the CODE-15%, ELSA-Brasil and SaMi-Trop cohorts, we computed the R square metric using linear regression and calculated the mean absolute error (MAE) using the chronological age. For further analysis, we divided the samples in three groups, based on differences between predicted ECG-age and chronological age: those with ECG-age more than 8 years smaller than the chronological age, those with ECG-age within a range of 8 years from their chronological age, and those ECG-age more than 8 years greater than the chronological age.

For mortality analysis, we used Cox proportional regression model, reporting hazard ratios (HR) and 95% confidence intervals (95% CI). The analysis was performed in all ECGs of the three cohorts, with two levels of adjustments: age and sex; age, sex, and other cardiac risk factors (hypertension, diabetes mellitus, and smoking). Other two models in the second level of adjustment including dyslipidemia, for CODE-15% and ELSA-Brasil, and obesity, only for ELSA-Brasil, were fitted. A second mortality analysis with the same parameters, was performed considering only normal ECGs from CODE-15% ($n = 80679$) and ELSA-Brasil ($n = 7691$) cohorts. The proportional hazard assumption was verified using a log ($-$log (survival)) plot and Schoenfeld residuals. We also performed the mortality analysis for CODE-15%, dividing the samples into five groups according to

quintiles of the difference of ECG-age and chronological age, showing the adjusted survival curves and HRs from the adjusted Cox models by age and sex. The area under the receiver operating characteristic curve (AUC) was used to evaluate the Cox model performance for 1-year mortality risk prediction.

To explore the association of risk factors with the ECG-age being more than 8 years greater than the chronological age we performed a logistic regression analysis for the ELSA-Brasil cohort including all ECG and only subjects with normal ECG. In this analysis we fitted a model for each risk factor adjusted by age and sex and reported the ORs and 95% confidence intervals.

**Reporting summary**. Further information on research design is available in the Nature Research Reporting Summary linked to this article.

## Data availability

SaMi-Trop cohort was made openly available (https://doi.org/10.5281/zenodo.4905618). The CODE-15% cohort was also made openly available (https://doi.org/10.5281/zenodo.4916206) The datasets contain information about mortality, age, sex, the ECG tracings, and the flag indicating whether the ECG tracing is normal. The DNN model parameters that give the results presented in this paper are also available (https://doi.org/10.5281/zenodo.4892365). This should allow the reader to partially reproduce the results presented in the paper. Restrictions apply to additional clinical information on the CODE-15% and SaMi-Trop cohorts; to the full CODE cohort; and, to the ELSA-Brasil cohort. Researchers affiliated to educational or research institutions might make requests to access the datasets. Requests should be made to the corresponding author of this paper. They will be forwarded and considered on an individual basis by the Telehealth Network of Minas Gerais and by ELSA-Brasil Steering Committee. An estimate for the time needed for data access requests to be evaluated is three months. If approved, any data use will be restricted to non-commercial research purposes. The data will only be made available on the execution of appropriate data use agreements.

## Code availability

The code for the model training, evaluation and statistical analysis is available at the github repository https://github.com/antonior92/ecg-age-prediction (the release at the time of submission was archived in https://doi.org/10.5281/zenodo.4975439 [43]).

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

## Acknowledgements

This research was partly supported by the Brazilian Agencies CNPq, CAPES, and FAPEMIG, by projects IATS, INCT-Cyber, MASWEB, and Atmosphere, by the Wallenberg AI, Autonomous Systems and Software Program (WASP) funded by Knut and Alice Wallenberg Foundation, and by the *Kjell och Märta Beijer Foundation*. The ELSA-Brasil study was supported by the Brazilian Ministries of Health and of Science and Technology (grants 01060010.00RS, 01060212.00BA, 01060300.00ES, 01060278.00MG, 01060115.00SP, and 01060071.00RJ). The SaMi-Trop cohort study is supported by the National Institutes of Health (P50 AI098461-02 and U19AI098461-06). A.H.R., B.B.D., P.A.L., S.M.B., L.G., W.M. and A.L.R. are recipients of unrestricted research scholarships from CNPq; E.M.S. and A.H.R. received scholarships from CAPES and CNPq; and D.M.O., W.M. and A.L.P.R. received a Google Latin America Research Award scholarship. We also thank NVIDIA for awarding our project with a Titan V GPU.

## Author contributions

E.M.L., G.M.M.P., A.H.R., T.B.S. and A.L.R. were responsible for the study design. A.L.R. conceived the project and acted as the project leader. A.H.R. choosed the neural network architecture, implemented, and tuned the deep neural network. E.M.L did the survival analysis and all the statistical tests. G.M.M.P. and A.L.R. interpreted the results and provided clinical interpretation. A.H.R., D.M.O. and P.R.G. were responsible for preprocessing the training data. P.R.G. was responsible for maintaining and extracting the CODE database. M.M.P.F., E.C.S., S.M.B., L.G. and B.B.D. were responsible for cohort design and management, data acquisition, follow-up, and ECG exams in ELSA-Brasil and Sami-Trop cohorts. G.M.M.P., A.H.R., E.M.L., W.M., T.B.S. and A.L.R. contributed to the writing and all authors revised it critically for important intellectual content. All authors read and approved the submitted manuscript.

## Funding

## Competing interests

The authors declare no competing interests.
