## [Peer Review File · Nature Communications]

Reviewers' Comments:

Reviewer #1:

Remarks to the Author:

The study of Lima EM et al. is considered novel and timely. Few comments for consideration:

- 1) The clinical application is not yet entirely clear. How would this influence interpretation of ECGs in, e.g., general practice, in the hospital, or for tele-visitation.
- 2) The first two sections in the discussion could preferably be included at an early stage of the manuscript to add understanding of the concept of ECG-age at an early stage.

Reviewer #2:

Remarks to the Author:

This paper analyzes the relationship between deep neural network predicted ECG-age and mortality on CODE-15%, ELSA-Brasil, SaMi-Trop datasets. The deep neural network is a deep residual connected convolutional neural network with batch normalization, ReLU activation, Dropout, and MaxPooling, modified based on the authors' previous work. Predicted ECG-age is derived from the trained model. Patients are divided into three based on differences between predicted ECG-age and chronological age with 8 years as a threshold. Results show that patients with predicted ECG-age more than 8 years greater than chronological age had a higher mortality rate. Overall, the paper is well-organized and easy to follow. It is important to see the predicted ECG-age as a biomarker for general mortality.

Detailed comments:

- Several concepts. The paper relates to several different but easily-confused concepts: "ECG-age", "chronological age", "predicted ECG-age", and "predicted chronological age". The paper aims to analyze the relationship between "predicted ECG-age" and "chronological age", and uses the predictions of the deep neural network as "predicted ECG-age". However, the deep neural network is actually trained with "chronological age", then the predictions should be called "predicted chronological age". To understand this, my guess is that there is an implicit assumption that the populational expectation of "ECG-age" is the same as "chronological age", so that one can use "predicted chronological age" to approximate "predicted ECG-age". If it is the case, it would be better to clearly list the above easily-confused concepts and illustrate the assumption.

- Model training. The paper says "The learning rate is reduced by a factor of 10 whenever the validation loss does not present any improvement for 7 consecutive epochs" while "The training runs for 70 epochs with the final model being the one with the best validation results during the optimization process", it is not clear whether learning rate decay meets 70 epochs in the training process. Since the mean square error (MAE) objective for age prediction is different from the previous cross-entropy objective for the classification task. It would be better to also see the learning curve during the training process.

- The paper chooses "8 years" as a threshold to divide people into 3 groups, because "The MAE in the CODE dataset is approximately 8 years". However, it would also be more interesting to see the fine-grained relationship between "differences of ECG age and chronological age" and outcomes firstly.

Questions:

- What are the details of age groups for hold-out test set splitting?
- What are the details of generating ECG tracings for saliency maps? E.g., any smoothing techniques post-processed on generated ECG tracings?

Reviewer #3:

Remarks to the Author:

The paper develops a deep learning model to predict age from ECGs and investigates its association with risk of death. The study is carried out at a massive scale with replication across multiple datasets. The paper has a few typos and clarity issue that I noted below, however there's a pair of issues that need closer attention.

- The paper says "This model indicates that participants with an estimated ECG-age of 8 or more years greater than the chronological age had higher mortality risk (HR 1.79, 95%CI 1.69-1.90; $p < 0.001$)." However when I look at Table 2, I see ECG-age > age+8y has the hazard ratio 0.78. The text looks swapped from the main paper. I can't tell which one is correct. There is a difference between the ages of the two groups (supp table 1). What could be happening is that predictions are low for old people and high for young people, so the residuals separate the old and young people and would have the hazards in Table 2. This might persist even with age correction if the correction is linear

- If the previous issue is just a typo, the discussion should have a limitation on the type of adjustment that was done. Given there's some difference in age and the covariates, I'd say the corrections were only "linear" and a non-linear correction with say a deep survival analysis model might reduce the observed relationship.

Minor:

Table 3 caption typo *under parenthesis"

Page 29 "thesplit" -> "the split"

Validation in the methods part should be described more carefully. How was the data split?

We are grateful to the reviewers for their precious comments and suggestions. We have studied the comments carefully and the manuscript has been modified attempting to address these. Additions and modifications that we think to be relevant to the reviewers were highlighted in **blue** in the new version of the manuscript.

We answer the reviewer's comments below.

Reviewer #1 (Remarks to the Author):

The study of Lima EM et al. is considered novel and timely. Few comments for consideration:

Thanks for the positive comments and for the suggestions. We believe both of them were very useful in improving the quality of the paper and below we describe how they were incorporated into the manuscript.

1) The clinical application is not yet entirely clear. How would this influence interpretation of ECGs in, e.g., general practice, in the hospital, or for tele-visitation.

Thank you for the question. When demonstrating that a 12-lead ECG could inform on future cardiovascular risk even after accounting for traditional cardiovascular factors, the information could be of help in decision making, especially in primary care. Recognizing potential risk modifiers can help doctors and patients in shared decision-making. We believe this paper is a first step in such a direction.

That is an important point and we included the following paragraph in the manuscript trying to make this clearer.

"...Our study is a further step towards a more practical use of the ECG in prognostic evaluation, considering that ECG-age can be a marker of the biological age of the cardiovascular system, or "cardiovascular age". This concept was introduced in previous studies^{27,28} with the purpose of improving risk communication and patient adherence to proposed interventions. There, however, the value does not provide additional information to what the calculated risk already informs (since is calculated based only on them). Doctors often struggle in decision making regarding treatment for primary prevention of cardiovascular diseases in intermediate risk patients. Identifying new risk modifiers that can potentially improve risk prediction in this population (either by a positive net reclassification index or by derivation of a new predictive model) is paramount. This is specially true if such a marker is derived from an inexpensive and widely available tool as the ECG. The analysis presented here shows that the

ECG-age can inform on risk that is not accounted for in traditional cardiovascular risk factors. And, in this sense, the concept can go beyond the concept of "cardiovascular age" proposed in previous studies^{27,28}. "

2) The first two sections in the discussion could preferably be included at an early stage of the manuscript to add understanding of the concept of ECG-age at an early stage.

Thank you for this suggestion. The two first paragraphs of the discussion were moved to the introduction to make it more clear. In order for them to fit well into the introduction and in the logic sequence we have inverted the order they appear so it now reads as follows:

"...This suggests that end-to-end models might be able to identify additional markers, that, in their turn, might be practical and useful tools in cardiovascular disease prediction.

In this context, we turn to the use of machine learning algorithms to infer age from ECG tracings^{11,12}. Previous studies have shown that the age estimated from the ECG (the ECG-age) is related to cardiovascular health^{11,12}: The ECG-age, calculated using a Bayesian model in 5-minutes ECGs, tended to be close to the chronological age in healthy non-athletes, whereas most subjects with risk factors or proven heart diseases had a predicted ECG-age that was higher than their chronological age¹¹; in another study, patients with a DNN-predicted age that exceeded the chronologic age by 7 or more years presented a higher frequency of low ejection fraction, hypertension, and coronary disease¹². More recently, ECG-derived age has also been related to vascular aging, measured by peripheral endothelial dysfunction¹³.

Biological aging refers to the decline in tissue/organismal function, whereas chronological aging simply indicates the time passed since birth¹⁴. In normally aging individuals, chronological and biological ages are the same. Biological aging, however, is affected by lifestyle, environmental factors, inheritable and acquired conditions, and diseases. Accelerated biological aging points to the decline of tissue/organismal function at a faster rate than the average, and hence associated with the risk of a premature death¹⁴. Most available biomarkers of biological age measure a specific aspect of aging, like molecular and cellular biomarkers, and functional and structural vascular parameters¹⁴. Multiple exams and composite scores are often needed, adding cost, risk, and complexity to the evaluation. ECG exams are low-cost and widely available, being part of

the routine evaluation of many patients in both primary and specialized care. Thus, if ECG can provide an accurate estimate of the biological age it can be potentially useful in clinical practise. "

Reviewer #2 (Remarks to the Author):

This paper analyzes the relationship between deep neural network predicted ECG-age and mortality on CODE-15%, ELSA-Brasil, SaMi-Trop datasets. The deep neural network is a deep residual connected convolutional neural network with batch normalization, ReLU activation, Dropout, and MaxPooling, modified based on the authors' previous work. Predicted ECG-age is derived from the trained model. Patients are divided into three based on differences between predicted ECG-age and chronological age with 8 years as a threshold. Results show that patients with predicted ECG-age more than 8 years greater than chronological age had a higher mortality rate. Overall, the paper is well-organized and easy to follow. It is important to see the predicted ECG-age as a biomarker for general mortality.

Thank you very much for the positive comments and the careful reading of our paper. We believe you raised important concerns, which we tried to address below. We believe this has contributed to improve the quality and clarity of our article and we hope it can now meet your expectations.

Detailed comments:

- Several concepts. The paper relates to several different but easily-confused concepts: "ECG-age", "chronological age", "predicted ECG-age", and "predicted chronological age". The paper aims to analyze the relationship between "predicted ECG-age" and "chronological age", and uses the predictions of the deep neural network as "predicted ECG-age". However, the deep neural network is actually trained with "chronological age", then the predictions should be called "predicted chronological age". To understand this, my guess is that there is an implicit assumption that the populational expectation of "ECG-age" is the same as "chronological age", so that one can use "predicted chronological age" to approximate "predicted ECG-age". If it is the case, it would be better to clearly list the above easily-confused concepts and illustrate the assumption.

Thank you for the comment. We completely agree that these terms can easily lead to confusion. In the end, there are only two concepts: the *age* and the *predicted age* (the model output). The additional qualifiers, however, try to offer some extra intuition concerning the interpretation of these quantities.

We use "predicted ECG-age" instead of just "predicted age" to stress the fact that the age is predicted solely from the ECG, and hence this predicted age is expected to have learned to capture, on average, how aging affects the ECG exam. We use "predicted ECG-age" and "ECG-age" interchangeably meaning: "the age predicted by AI from the raw ECG tracing". In this sense, our view of the problem was that "ECG-age" is something we extract and not as some inherent characteristic of a given patient, or: we are implicitly assuming that every ECG-age is a prediction.

The word *chronological* was picked to emphasize the difference between the true patient age from the age that is predicted by a model. Concerning the possibility of using "*predicted chronological age*" to refer to the model prediction we are slightly afraid the modification would make the two terms we want to contrast even more similar and hence the text more difficult to read.

In an attempt to address the confusion between the terms we did the following:

- a. We made the nomenclature uniform throughout the text by using "*(predicted) ECG-age*" as the age predicted by AI from the raw ECG tracing and "*chronological age*" as the age the person has at the moment the exam was taken. Other word choices were removed to avoid confusion.
- b. We added a paragraph explaining the concept of "*(predicted) ECG-age*" and "*chronological age*" to the introduction:

"We build on the hypothesis that an AI model exposed to many ECG exams with the task of predicting the age might learn to capture, on average, how aging affects the ECG exam. Age is a risk factor for cardiac diseases that affects ECG measurements and the likelihood of having an ECG with a higher incidence of abnormalities^{16,17}. Hence, here we study the possibility of using the AI predicted ECG-age as an indicator of cardiovascular health. We refer to this age predicted by the AI model from the raw 12-lead ECG tracing as predicted ECG-age, or just ECG-age, and, to contrast that, we refer to the patient age as chronological age."

c. We also rewrote some parts of the paragraph in the discussion which reflects on these terms. Now it reads:

"In this paper, we use a data-driven approach to obtain a model that predicts age from the raw ECG tracing. By having the chronological age of the person as the prediction target, we expect the trained model to learn to capture, on average, how aging affects the ECG exam. Indeed, having a predicted ECG-age higher than one's actual age is an indication that the exam is similar to those of older people, who have a higher associated cardiovascular risk and are more likely to die from cardiovascular diseases. ..."

- Model training. The paper says "The learning rate is reduced by a factor of 10 whenever the validation loss does not present any improvement for 7 consecutive epochs" while "The training runs for 70 epochs with the final model being the one with the best validation results during the optimization process", it is not clear whether learning rate decay meets 70 epochs in the training process. Since the mean square error (MAE) objective for age prediction is different from the previous cross-entropy objective for the classification task. It would be better to also see the learning curve during the training process.

Good point. We included the learning curve as Supplementary Figure 3 in the manuscript. The learning rate starts at $1e-3$ and reaches $1e-7$ at the final stage of the training. The performance does not change that much after epoch 40 but the best model is obtained in epoch 66 with MAE=8.5097.

Sup Fig 3: Learning curve. In blue, the plot displays the mean absolute error (MAE) computed in the 5% validation set. In gray, it shows the learning rate used. The x-axis gives the epochs: each epoch a full pass through the set of training examples updating the model weights. The best model is obtained in epoch 66 with MAE=8.5097 on the validation set.

- The paper chooses “8 years” as a threshold to divide people into 3 groups, because “The MAE in the CODE dataset is approximately 8 years”. However, it would also be more interesting to see the fine-grained relationship between “differences of ECG age and chronological age” and outcomes firstly.

Thank you for your suggestion. We included a more fine grained analysis of the results in the manuscript. Supplementary Figure 4 contains the results of this analysis. There we display the adjusted survival and hazard ratio from adjusted Cox regression models for a variable defined in five groups according to quintiles of the difference of ECG-age and chronological age. The reference group is the one that includes the difference of ECG-age and chronological age equal to zero.

Sup Fig 4. Adjusted survival curve for patients quintiles. Adjusted survival curves, hazard ratios (HR) and CIs 95% for 5 groups of the difference between ECG-age and chronological age. The patients were divided into quintiles according to quintiles of the difference of ECG-age and chronological age. The HR summarizes the Cox regression models obtained for overall mortality. The models were adjusted by age and sex.

In the results, we included a section titled **"A fine-grained analysis of the relation between ECG-age and mortality"** describing this analysis.

"In most of the discussion in this paper we use a hard threshold of 8 years old between the predicted ECG-age and the chronological age to separate the patients into different risk groups. In Sup Fig 4, we present an alternative setup where we split the patients into 5 quintiles of the difference of predicted ECG-age and chronological age and show adjusted survival, hazard ratios and 95% confidence intervals for these groups. The results indicate that the groups where predicted ECG-age is lower than the chronological age (Q1 and Q2) had a lower risk of mortality and the groups

Q4 and Q5 had a higher risk of mortality. Both when all exams are considered and when only normal exams are considered."

Questions:

- What are the details of age groups for hold-out test set splitting?

A stratified sampling was used to generate the hold-out test set. That means we group the patients in 70 uniformly spaced age groups that range from 16 to 85 years. We then pick at random the same number of patients from each group. We rewrote the section "The CODE-15% cohort" in methods, trying to make it clearer and giving additional details about how the hold-out set was generated.

"The CODE-15% is a subset of the CODE cohort. The CODE cohort was divided into 85-15% splits, with the 85% split being used for developing the model and 15% hold-out being the one used in subsequent analyses and referred to as CODE-15%. This hold-out set is obtained by a stratified sampling procedure, where the stratification is made with respect to the patients age. The procedure is illustrated in Sup Fig 5. Given all the exams from the original CODE cohort, we group the exams by the age of the patient at the time of the examination. One group for each age ranging from 16 to 85 years, i.e. a total of 70 uniformly spaced age groups. The CODE-15% cohort is obtained by picking the same number of exams (~3100) at random from each age group. The result is an, approximately, uniform age distribution from 16 years to 85 years. Only the first patient exam is considered in all the analysis with this cohort and the remaining exams from the patients are removed from the remaining data and not used in the analysis. We do not sample from patients older than 85 or younger than 16 years, which do not appear in the CODE-15% cohort."

We also added an illustrative diagram explaining the stratified sampling:

Sup Fig 5: Illustrative representation of stratified sampling. We illustrate the stratified sampling used to generate the CODE-15% dataset. Each ECG exam is represented by a disk. The exams are divided into age groups with one group for each age, ranging from 16 to 85 years old. The same number of samples from each age group is then randomly picked to be assigned to the CODE-15% dataset. The same procedure is used to generate the 5% split used for validating the model.

- What are the details of generating ECG tracings for saliency maps? E.g., any smoothing techniques post-processed on generated ECG tracings?

There is no post-processing for generating the plot in **Sup Fig 1**. We use quite a "vanilla" technique for generating the saliency maps: For a given exam, we feed it to the deep neural network and compute the ECG-age. Using backpropagation we compute the derivative of the ECG-age with respect to each point. We then generate transparent blue disks in the same plot as the ECG-tracing, with the size of these disks proportional to the magnitude of the derivative in this point.

We rewrote the paragraph describing the procedure:

"The results are displayed in Sup Fig 1 and the relative size of the blue disks in the image might be interpreted as the relative importance of each point to the output prediction (at least in terms of the linearized local

analysis). Similar approaches have been pursued in the interpretation of other DNN-based ECG predictors^{8,42}. The saliency maps are generated using the following procedure²¹: the raw ECG tracing is fed to the deep neural network and the ECG-age is computed. Using backpropagation we compute the derivative of the ECG-age with respect to each point. We then generate transparent blue disks in the same plot as the ECG-tracing, where the size of these disks is proportional to the magnitude of the derivative in this point. This procedure results in the saliency map displayed in Sup Fig 1."

Since the first appearances of saliency maps for auxiliary the interpretation of deep neural networks there have been several developments, i.e. grad-CAM is an example of popular choice (Selvaraju et. al., 2020). And, indeed, many of these do have more sophisticated post-processing stages. Many of these developments, however, are for classification problems and try to address the challenges of visualizing the relative importance of features in a multiclass setting. Since our problem is a regression, we opt for the simple, yet well fundamented, implementation described in (Simonyan, 2013), which does not include additional post-processing steps.

We included the following footnote to avoid confusion:

"Here we use a rather straightforward procedure for the generation of saliency maps, which also was one of the first methods to be proposed²¹. More sophisticated versions have appeared later, such as Grad-CAM⁴³, which are popular for classification problems and have been used for ECG interpretation before^{8,42}. These, however, are for classification problems and try to address the challenges of visualizing the relative importance of features in a multiclass setting. Since we have a regression problem many of these are not directly applicable. Here we opt for the simple, yet well motivated, implementation²¹ we described above."

References:

R. R. Selvaraju, M. Cogswell, A. Das, R. Vedantam, D. Parikh, and D. Batra, "Grad-CAM: Visual Explanations from Deep Networks via Gradient-based Localization," *Int J Comput Vis*, vol. 128, no. 2, pp. 336–359, Feb. 2020, doi: [10.1007/s11263-019-01228-7](https://doi.org/10.1007/s11263-019-01228-7).

Simonyan, K., Vedaldi, A. & Zisserman, A. Deep Inside Convolutional Networks: Visualising Image Classification Models and Saliency Maps. *arXiv:1312.6034* (2013).

Reviewer #3 (Remarks to the Author):

The paper develops a deep learning model to predict age from ECGs and investigates its association with risk of death. The study is carried out at a massive scale with replication across multiple datasets. The paper has a few typos and clarity issue that I noted below, however there's a pair of issues that need closer attention.

Thank you very much for the thoughtful comments. We believe you touch upon some very important and interesting ideas and we have tried our best to address the issues you have raised. We believe this has contributed to improve the quality and clarity of our article and we hope it can now meet your expectations.

- The paper says "This model indicates that participants with an estimated ECG-age of 8 or more years greater than the chronological age had higher mortality risk (HR 1.79, 95%CI 1.69-1.90; $p < 0.001$)." However when I look at Table 2, I see ECG-age > age+8y has the hazard ratio 0.78. The text looks swapped from the main paper. I can't tell which one is correct.

We thank the reviewer for finding this error and we are deeply sorry for it. The table results were swapped. This has been updated in the new version of the manuscript.

There is a difference between the ages of the two groups (supp table 1). What could be happening is that predictions are low for old people and high for young people, so the residuals separate the old and young people and would have the hazards in Table 2. This might persist even with age correction if the correction is linear

We thank the reviewer for the comment. We note that the concern with the hazards in Table 2 is addressed by the correction of the previous point (the table results were switched). We stress that this confusion is due to our mistake in the table, and that the reviewers' point was coherent with the first manuscript.

Second, we observe that assuming that ECG Age is indeed a predictor of mortality, we would expect individuals that have an ECG-age higher than their chronological age would be more likely to die, and therefore, not be a part of the sample we are observing to begin with. This observation can help to interpret Sup table 1, it explains the counter-intuitive finding that people with ECG ages < age-8y are usually older than those with ECG ages > age+8y. To make this point clear, we added the following to the text:

"Sup Table 1 presents baseline characteristics for each of the cohorts by ECG-age groups. We find that in all three cohorts, the group of patients with an ECG-age more than 8 years greater than the chronological age have the lowest average chronological age (CODE-15%: 42.2;

ELSA-Brasil: 48.5; SaMi-Trop: 54, $p < 0.001$ for t-tests comparing average ages with the other ECG-age strata). Although seemingly contradictory, this is in accordance with the hypothesis that ECG-age is indeed a predictor of mortality. Since patients that have an ECG-age higher than their chronological age are more likely to die, older patients whose ECG-age is higher than their chronological age are not likely to be a part of the sample we are analyzing.”

- If the previous issue is just a typo, the discussion should have a limitation on the type of adjustment that was done. Given there's some difference in age and the covariates, I'd say the corrections were only "linear" and a non-linear correction with say a deep survival analysis model might reduce the observed relationship.

We agree that this is an important point and we added the following in the discussion trying to address it:

“Here, we present the mortality risk prediction as a downstream task. That is, a model that was trained for predicting the patient's age is later used for a different task: that of mortality prediction. This shows the model is useful in scenarios that it has not been explicitly trained on and when used in a simple linear cox model it can help separate patients in different risk groups. Nonetheless, one possible limitation of this analysis is that the relations considered in this second step are only the linear ones. Hence, taking into consideration nonlinear relations in this second step could possibly modify the observed relationship.”

Minor:

Table 3 caption typo *under parenthesis"

Page 29 "thesplit" -> "the split"

We fixed both typos. Thank you for spotting them.

Validation in the methods part should be described more carefully. How was the data split?

We used a 80%-5%-15% split in the code cohort. CODE-15% is used as a hold out test set together with the two other sets (ELSA-Brasil and SaMi-Trop). The 5% split is used for validation and the 80% is used for training. We use stratified sampling to guarantee an approximately uniform age distribution in the test and validation sets.

The stratified sampling procedure is illustrated in the figure below:

Sup Fig 5: Illustrative representation of stratified sampling. We illustrate the stratified sampling used to generate the CODE-15% dataset. Each ECG exam is represented by a disk. The exams are divided into age groups with one group from each age from 16 to 85 years old. The same number of samples from each age group is then randomly picked to be assigned to the CODE-15% dataset. The same procedure is used to generate the 5% split used for validating the model.

In the new version of the manuscript, we also try to describe the data split more carefully. Section "**The CODE-15% cohort**", describing the construction of the 15% hold out test set used for testing the model was rewritten:

"The CODE-15% is a subset of the CODE cohort. The CODE cohort was divided into 85-15% splits, with the 85% split being used for developing the model and 15% hold-out being the one used in subsequent analyses and referred to as CODE-15%. This hold-out set is obtained by a stratified sampling procedure, stratified by the patients age. The procedure is illustrated in Sup Fig 4. Given all the exams from the original CODE cohort, we group the exams by the age of the patient at the moment it was taken. One group for each age ranging from 16 to 85 years, i.e. a total of 70 uniformly spaced age groups. The CODE-15% cohort is obtained by picking the same number of exams (~3100) at random from each age group. The result is an, approximately, uniform age distribution from years 16 to 85 years. Only the first patient exam is considered in all the analysis with this

cohort and the remaining exams from the patients are removed from remaining data and not used in the analysis. We do not sample from patients with more than 85 or less than 16 which do not appear in the CODE-15%."

And, in the section "**Model**", in methods, the description of the split between training and validation data was also rewritten:

"Exams from patients in the CODE cohort that were not included in the hold-out set CODE-15% (see section above) were used to develop a convolutional DNN to predict age. This split contains 85% of the patients and was further divided into 80-5% splits: being the first used to learn the neural network weights, and the samples from the 5% remaining patients used for comparing design choices and adjusting optimization parameters.

The 5% validation split for is obtained by using a stratified sampling procedure. The procedure is illustrated in Sup Fig. 4 and is similar to the one used for generating the CODE-15% cohort. Given all the patients that are not in the CODE-15% cohort, we group their exams by the age of the patient at the moment it was taken. One group for each age ranging from 16 to 85 years. The 5% validation set is then obtained by sampling the same number of exams (~1600) at random from each age group. As for the CODE-15% cohort, such procedure aims to guarantee an, approximately, uniform age distribution in the validation set, by picking the same number of patient exams for equally-spaced one-year intervals. The training dataset is composed of all exams from the 80% remaining patients. ..."

Reviewers' Comments:

Reviewer #2:

Remarks to the Author:

Thanks for your responses. I have no further concerns.

Reviewer #3:

Remarks to the Author:

I appreciate the author's response to my main issue and enjoyed the updated manuscript.

REVIEWERS' COMMENTS

Reviewer #2 (Remarks to the Author):

Thanks for your responses. I have no further concerns.

Reviewer #3 (Remarks to the Author):

I appreciate the author's response to my main issue and enjoyed the updated manuscript.

Response

We thank the reviewers for providing helpful feedback and useful suggestions.